# Language Models are Symbolic Learners in Arithmetic

**Chunyuan Deng**                                    *chunyuan.deng@rice.edu*
*Department of Computer Science*
*Rice University*

**Zhiqi Li**                                              *zli3167@gatech.edu*
*College of Computing*
*Georgia Institute of Technology*

**Roy Xie**                                              *ruoyu.xie@duke.edu*
*Department of Computer Science*
*Duke University*

**Ruidi Chang**                                          *rc151@rice.edu*
*Department of Computer Science*
*Rice University*

**Hanjie Chen**                                          *hanjie@rice.edu*
*Department of Computer Science and Ken Kennedy Institute*
*Rice University*

**Reviewed on OpenReview:** *https://openreview.net/forum?id=QSblPg1xUM*

## Abstract

The prevailing question in LM performing arithmetic is whether these models learn to truly compute or if they simply master superficial pattern matching. In this paper, we argues for the latter, presenting evidence that LMs act as greedy symbolic learners, prioritizing the simplest possible shortcuts to fit the stats of dataset to solve arithmetic tasks. To investigate this, we introduce **subgroup induction**, a practical framework adapted from Solomonoff Induction (SI), one of the most powerful universal predictors. Our framework analyzes arithmetic problems by breaking them down into "subgroups"—minimal mappings between a few input digits and a single output digit. Our primary metric, subgroup quality, measures the viability of these shortcuts. Experiments reveal a distinct U-shaped accuracy pattern in multi-digit multiplication: LMs quickly master the first and last output digits while struggling with those in the middle. We demonstrate this U-shape is not coincidental; it perfectly mirrors the quality of the simplest possible subgroups, those requiring the fewest input tokens. This alignment suggests a core learning mechanism: LMs first learn easy, low-token shortcuts and only incorporate more complex, multi-token patterns as training progresses. They do not learn the algorithm of multiplication but rather a hierarchy of increasingly complex symbol-to-symbol mappings. Ultimately, our findings suggest that the path to arithmetic mastery for LMs is not paved with algorithms, but with a cascade of simple, hierarchically-learned symbolic shortcuts. The code is at https://github.com/chili-lab/Symbolic-Arithmetic.

## Introduction

The saturation of modern math benchmarks like GSM8K (Cobbe et al., 2021) by frontier language models such as GPT-4o (OpenAI et al., 2024) and Claude (Anthropic, 2024) marks a significant milestone in AI. As the research community pivots to grander challenges like Olympic-level mathematics, this success raises a more fundamental question: are these models developing genuine numerical reasoning, or are they exhibiting

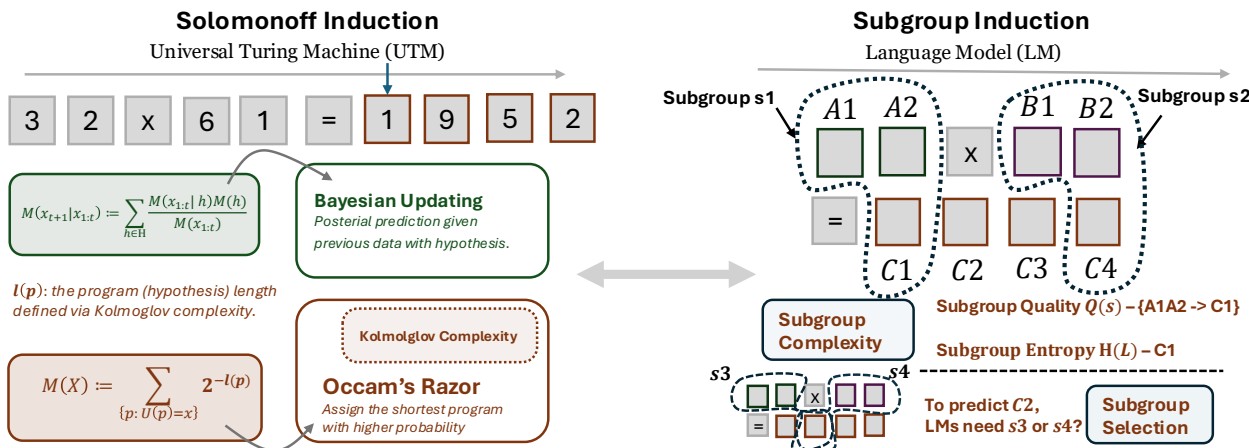

Figure 1: **Overview of Subgroup Induction.** Solomonoff Induction (SI) is a conceptual framework that utilizes a universal predictor, such as UTMs, to make predictions. Inspired by SI's principle of Occam's Razor, subgroup induction is a pragmatic framework designed to uncover the shortcut-seeking mechanisms LMs use to perform arithmetic.

an elaborate illusion of it, woven from the statistical patterns of their vast training data? In short, do they learn to truly *compute*, or do they simply master sophisticated pattern matching?

In this paper, we argue for the latter. We present compelling evidence that LMs act as **greedy symbolic learners**, prioritizing the simplest possible "shortcuts" to solve arithmetic tasks. Understanding this distinction is critical, as it has profound implications for model reliability, out-of-distribution generalization, and the future of trustworthy AI reasoning. To probe this behavior, we use arithmetic as a controlled laboratory. Unlike open-ended reasoning, arithmetic tasks are **close-formed**—the underlying data generation function $f$ is pre-defined and immutable. This provides a gold-standard ground truth against which we can rigorously analyze the learning strategies LMs actually develop.

To formalize this investigation, we introduce a practical framework called **subgroup induction** (see Figure 1). Our approach is inspired by one of the oldest principles in science and learning theory: Occam's Razor, which suggests a preference for simpler explanations. This principle is formally embodied in theoretical frameworks like Solomonoff Induction (SI). We adapt this core idea into a concrete, testable hypothesis for LMs: a "simpler" solution for predicting an output digit is one that relies on the simplest possible input pattern—that is, a minimal set of input digits (tokens).

Within this framework, a **subgroup** becomes our basic unit of analysis. It represents a potential shortcut, defined as a minimal mapping from a subset of input digits to a single output digit. For example, does an LM only need to see the last digits '...3' and '...4' to learn that the product must end in '...2'? To measure the viability of such shortcuts, we introduce two complementary measures of **subgroup complexity**. Our primary metric, *subgroup quality*, quantifies the reliability of a given shortcut across the entire data distribution. A second metric, *subgroup entropy*, measures the inherent ambiguity or difficulty of the prediction at each output position, serving as a robust tool for error estimation.

Our experiments reveal a distinct and consistent U-shaped accuracy pattern in multi-digit multiplication: LMs rapidly master the first and last output digits but consistently struggle with those in the middle. We demonstrate this U-shape is not coincidental but is the "smoking gun" for our shortcut hypothesis. The model's performance curve perfectly mirrors the quality of the simplest possible subgroups—those requiring **the fewest input tokens**. This powerful alignment reveals a core learning mechanism: *LMs instinctively learn easy, low-token shortcuts first and only incorporate more complex, multi-token patterns when forced to by the learning objective.* They do not learn the abstract algorithm of multiplication; instead, they discover a hierarchy of symbol-to-symbol mappings, starting with the path of least resistance.

Furthermore, we show that subgroup entropy is a powerful tool for analyzing more complex reasoning. In a Chain-of-Thought (CoT) setting (Wei et al., 2023), different reasoning paths can be viewed as decompositions

of a problem into simpler steps. Our framework quantifies the "symbolic complexity" of each path, and we find that the CoT path with the lowest aggregate entropy consistently achieves the best performance. This suggests LMs prefer reasoning strategies composed of the easiest possible sequence of symbolic shortcuts.

# 1  Preliminaries

In this section, we introduce the preliminaries of the universal Turing machine and Solomonoff induction. We then discuss how arithmetic learning can be incorporated into the SI framework, laying the groundwork for presenting our subgroup induction framework .

## 1.1  Solomonoff Induction

Solomonoff induction formalizes optimal prediction by combining computability and Bayesian principles. It considers all possible hypotheses (as programs) weighted by two criteria: simplicity (shorter programs receive exponentially higher weight) and consistency with observed data.

The framework unifies three elements: a universal prior $\mathbf{M}$ over hypotheses, Kolmogorov complexity as a measure of simplicity, and Bayesian updating across the hypothesis space. This yields provably optimal predictions for sequences generated by any computable process. We formalize these components as follows:

**Definition 1.1** (Universal Probability)**.** The universal probability $\mathbf{M}$ is a mixture of all lower semicomputable semimeasures $\mu$, weighted by their algorithmic complexity:

$$\mathbf{M}(x) = \sum_{p:U(p)=x} 2^{-l(p)}$$

where $l(p)$ is the length of program $p$ on the reference UTM $U$. Each program $p$ represents a hypothesis about the data-generating process.

**Definition 1.2** (Occam's Razor)**.** The principle of Occam's Razor is formalized through Kolmogorov complexity $K_U(x)$, which is the length of the shortest program that outputs $x$:

$$K_U(x) = \min\{l(p) : U(p) = x\}$$

This assigns higher prior probabilities $(2^{-K_U(x)})$ to simpler hypotheses, as shorter programs have higher weights in the universal probability.

**Definition 1.3** (Bayesian Updating)**.** Each program $p$ maps to a hypothesis $h$ in hypothesis space $\mathcal{H}$. Given prior probability $\mathbf{M}$ and observation $x_{1:t}$, the posterior probability is computed via Bayes' rule:

$$\mathbf{M}(h|x_{1:t}) = \frac{\mathbf{M}(x_{1:t}|h)\mathbf{M}(h)}{\mathbf{M}(x_{1:t})}$$

The predictor updates across all hypotheses (programs):

$$\mathbf{M}(x_{t+1}|x_{1:t}) = \sum_{h \in \mathcal{H}} \mathbf{M}(x_{t+1}|h)\mathbf{M}(h|x_{1:t})$$

## 1.2  Arithmetic Learning as a Bridge to Practice

Arithmetic learning offers an ideal setting for studying inductive inference, as it involves ground-truth data-generating functions. We will first provide a formal definition of arithmetic learning and then discuss the similarities that motivate the development of a practical framework.

**Arithmetic Learning Task.**  The task of arithmetic learning to learn an approximation to ground truth data generation function $\hat{f}$ that generalizes to unseen inputs. The ground truth $f : \mathbb{N} \times \mathbb{N} \to \mathbb{N}$ maps input pairs to their arithmetic result:

$$f(a, b) = a \diamond b = c$$

where $\diamond$ represents an arithmetic operation like addition or multiplication.

**Arithmetic Data Distribution.** Consider a training set $\mathcal{D}_{\text{train}} = \{(a^{(k)}, b^{(k)}, c^{(k)})\}_{k=1}^N$ where $c^{(k)} = f(a^{(k)}, b^{(k)})$ for a binary operator $f(\cdot)$. For $n$-digit arithmetic, inputs $a^{(k)}$ and $b^{(k)}$ can be viewed as realizations of random variable sequences $\{A_i\}_{i=1}^n$ and $\{B_i\}_{i=1}^n$, where each $A_i, B_i \sim \mathcal{U}\{0, 1, \ldots, 9\}$, $\mathcal{U}$ is the uniform distribution. Similarly, $c^{(k)}$ corresponds to sequence $\{C_i\}_{i=1}^m$ with joint distribution:

$$P(\{C_i\}_{i=1}^m, \{A_i\}_{i=1}^n, \{B_i\}_{i=1}^n) = I_{\{f(a,b)=c\}} P(A) P(B)$$

where $I_{\{\cdot\}}$ is the indicator function.

**LMs as Predictive Distributions.** LMs implement a distribution $p_\theta(x_{t+1}|x_{1:t})$ over next tokens, parameterized by weights $\theta$. For arithmetic learning:

$$p_\theta(c|a, b) \approx \mathbb{P}(f(a, b)|a, b)$$

This approximates the true conditional probability of the arithmetic result.

**Proposition 1.4** (Parallel Learning Frameworks). *The theoretical SI framework and practical LM implementation approach arithmetic learning through parallel mechanisms: 1. SI via Bayesian updating over programs:*

$$\mathbf{M}(c|a, b, \mathcal{D}) = \sum_{h \in \mathcal{H}} \mathbf{M}(c|h) \mathbf{M}(h|\mathcal{D})$$

*2. LMs via gradient descent on negative log-likelihood:*

$$\mathcal{L}(\theta) = - \sum_{(a,b,c) \in \mathcal{D}} \log p_\theta(c|a, b)$$

Both frameworks exhibit convergence behaviors on arithmetic tasks: 1. SI identifies the shortest program computing $f$ with probability 1 as $|\mathcal{D}| \to \infty$ 2. LMs learn an approximate implementation minimizing empirical risk, bounded by model capacity and optimization dynamics.

These comparisons demonstrate that it is possible to use the core principles of Solomonoff induction to understand the behavior of language models in arithmetic tasks. This insight motivates us to develop a pragmatic framework capable of providing computable properties to analyze LMs' learning mechanisms in arithmetic.

## 2 Subgroup Induction

In this section, we introduce the subgroup induction framework. To constrain the hypothesis space, we define a computable property by specifying the concept of a subgroup.

### 2.1 Subgroup

**Definition 2.1** (Subgroup). For $n$-digit arithmetic $f(a, b) = c$, the subgroup $s$ is defined as ($A_i$, $B_i$, and $C_i$ have the same meaning as in subsection 1.2):

$$s \in \mathbb{S}_n = \{((\mathbb{A}, \mathbb{B}), \mathbb{C}) \mid \mathbb{A} \subseteq \{A_i\}_{i=1}^n, \mathbb{B} \subseteq \{B_i\}_{i=1}^n, \mathbb{C} \in \{C_1, C_2, \ldots, C_m\}\}$$

where $\mathbb{A}$ and $\mathbb{B}$ are subsets of the input variable sequence, $\mathbb{C}$ represents a single variable in the output sequence, $n$ and $m$ is the length for input and output number. For convenience, we represent $\mathbb{A}, \mathbb{B}, \mathbb{C}$ by the sequence of their elements in order. For example, if $\mathbb{A} = \{A_1, A_3, A_5\}$, we write $\mathbb{A}$ as $A_1 A_3 A_5$. Examples for 2-digits multiplication $A_1 A_2 \times B_1 B_2 = C_1 C_2 C_3 C_4$:

- $((A_1, B_1), C_1)$
- $((A_2, B_2), C_4)$
- $((A_1 A_2, B_1), C_1)$

- $((A_1, B_1 B_2), C_4)$
- $((A_1 A_2, B_1 B_2), C_4)$
- ...

In these subgroups, $\mathbb{A}$ *and* $\mathbb{B}$ *are subsets of input variable sequences, while* $\mathbb{C}$ *is restricted to a single variable.* This design aligns with the next-token prediction paradigm in language modeling.[1]

**Computability.** By using subgroup decomposition, arithmetic tasks with $n$-digit input and $m$-digits output can be divided into $(2^{2n} - 1) \times 2m$ subgroups overall.[2] Essentially, *we treat the LMs' choice of using symbol information, which corresponds to choosing a subgroup, as a hypothesis.*[3]

## 2.2 Subgroup Complexity

In Solomonoff Induction, hypotheses or universal turing machine programs are assigned a complexity measure. Kolmogorov complexity, a key component of this framework, embodies Occam's Razor by assigning higher probability to shorter programs. In our framework, we treat subgroup usage as a hypothesis. Thus, a key challenge is defining an appropriate measure for subgroup complexity.

**Intuition.** In multiplication, *the leftmost input digits largely determine the first output digits, hinting that predicting some positions might be easy.* Also, *predicting a single value* 0 *is easier than predicting a range* $0 - 9$, *showing that output variability matters.* These two observations motivate our two proposed complexity measures respectively.

**Definition 2.2** (Subgroup Quality). Given original dataset $\mathcal{D}$, for any subgroup $s = ((\mathbb{A}, \mathbb{B}), \mathbb{C}) \in S_n$, the subgroup quality for multiplication is:

$$Q(s) \;=\; \mathbb{E}_{(a,b,c) \sim D}\Big[\mathbf{I}\Big(\phi(a, \mathbb{A}) \,\times\, \phi(b, \mathbb{B}) = \phi(c, \mathbb{C})\Big)\Big] \tag{1}$$

where $\mathbf{I}(\cdot)$ is an indicator function returns 1 if argument is true, and 0 otherwise. $\phi$ is a masking function to set digits at positions of variables not in subgroup to 0 (see Algorithm 1).

Extending this definition from multiplication to any operator $f(\cdot)$, we could a generalized version:

$$Q(s) \;=\; \mathbb{E}_{(a,b,c) \sim D}\Big[\mathbf{I}\Big(f(\phi(a, \mathbb{A}), \phi(b, \mathbb{B})) = \phi(c, \mathbb{C})\Big)\Big] \tag{2}$$

where $f(\cdot)$ is any binary operator for the arithmetic tasks.

**Definition 2.3** (Subgroup Entropy). The subgroup space is defined based on the label space for each subgroup. Given $s$, the label space is given by:

$$\mathbb{L}_s = \{c \mid P(\mathbb{C} = c) > 0\}$$

For a given subgroup, the label space consists of all labels observed at its output positions in the dataset. *The subgroup entropy is then defined as*:

$$H(s) = - \sum_{c \in \mathbb{L}_s} P(\mathbb{C} = c) \log_2 P(\mathbb{C} = c) \tag{3}$$

## 2.3 Discussions

For a given subgroup $s$, subgroup complexity is defined using two computable measures:

---

[1]To prevent tokenization issues, we add a space between each digit to ensure that each digit is tokenized separately.

[2]Input and output each have $2n$ digits. Considering all possible permutations for input (excluding the empty set), we have $(2^{2n} - 1)$ combinations. Therefore, the total number of subgroups is $(2^{2n} - 1) \times 2m$.

[3]ML models like linear/logistic regression or simple MLP, *the hypothesis is characterized by the model weights (parameters)* that define the mapping function $\pi_\theta$. But this also face the challenge of incomputability due to the infinite hypothesis space.

---

**Algorithm 1** Subgroup Quality for $n$-digit Multiplication

---

**Require:** Original dataset $\mathcal{D}_{\text{train}}$ of size $N$, set of all subgroups $S$

    **Note:** MASK sets digits at positions of variables *not* in the subgroup to 0

1: **for** each subgroup $s = ((\mathbb{A}, \mathbb{B}), \mathbb{C}) \in S$ **do**
2:     $correct \leftarrow 0$
3:     **for** each $(a, b, c) \in \mathcal{D}_{\text{train}}$ **do**
4:         $a_{\text{masked}} \leftarrow \text{MASK}(a, \mathbb{A})$
5:         $b_{\text{masked}} \leftarrow \text{MASK}(b, \mathbb{B})$
6:         $c_{\text{masked}} \leftarrow \text{MASK}(c, \mathbb{C})$
7:         $pred \leftarrow a_{\text{masked}} \times b_{\text{masked}}$
8:         **if** $pred = c_{\text{masked}}$ **then**
9:             $correct \leftarrow correct + 1$
10:        **end if**
11:     **end for**
12:     $Q(s) \leftarrow correct/N$                                     ▷ subgroup quality
13: **end for**

---

- $Q(s)$: subgroup quality that measures how accurate the mapping is from the input positions to the output position.

- $H(s)$: Subgroup entropy that measures the uncertainty or variability within the subgroup's output labels.

These two measures are proposed with different motivations. Next, we will explore the effectiveness of these two measures in quantifying LMs' learning behavior.

## 3 Arithmetic Learning in Language Models

We investigate LMs' learning mechanism using an intuitive yet straightforward approach by observing the relationship between subgroup complexity and position-level accuracy.

### 3.1 Experiment Settings

We trained Gemma-2-2B (Team et al., 2024) and Llama-3.1-8B (Dubey et al., 2024) on multiplication arithmetic tasks using datasets of varying sizes: 6.48K, 12.96K, 32.4K, and 64.8K examples. Our experiments covered multiplication of 3 to 5-digit numbers, resulting in outputs ranging from 6 to 10 digits. For simplicity and to ensure a focused comparison, we did not optimize hyperparameters for either Gemma or Llama models, as preliminary experiments indicated robust performance with default settings. Detailed experimental setup information can be found in Appendix A.

### 3.2 Position-level Accuracy are U-shaped

We compute the accuracy at each position in the output sequence. Figure 2 reveals a phenomenon overlooked in previous studies that fundamentally challenges our understanding of how language models learn arithmetic operations. Contrary to the common assumption that position-level accuracy decreases monotonically from right to left due to carryover effects and least-to-most significant digit calculations (Lee et al., 2023; Zhang-Li et al., 2024), our results show a **U-shaped accuracy curve** in both Gemma-2-2B and Llama-3.1-8B models across all multiplication complexities.

This U-shaped pattern exhibits several noteworthy characteristics. Accuracy peaks at the beginning and end positions, consistently exceeding 95% regardless of training set size, while dramatically dropping to approximately 10% in the middle positions, particularly pronounced in higher-digit multiplication tasks. The consistency of this pattern across different model architectures (Gemma-2-2B and Llama-3.1-8B) and training set sizes (ranging from 6.48K to 64.8K examples) suggests that this is not an artifact of specific training

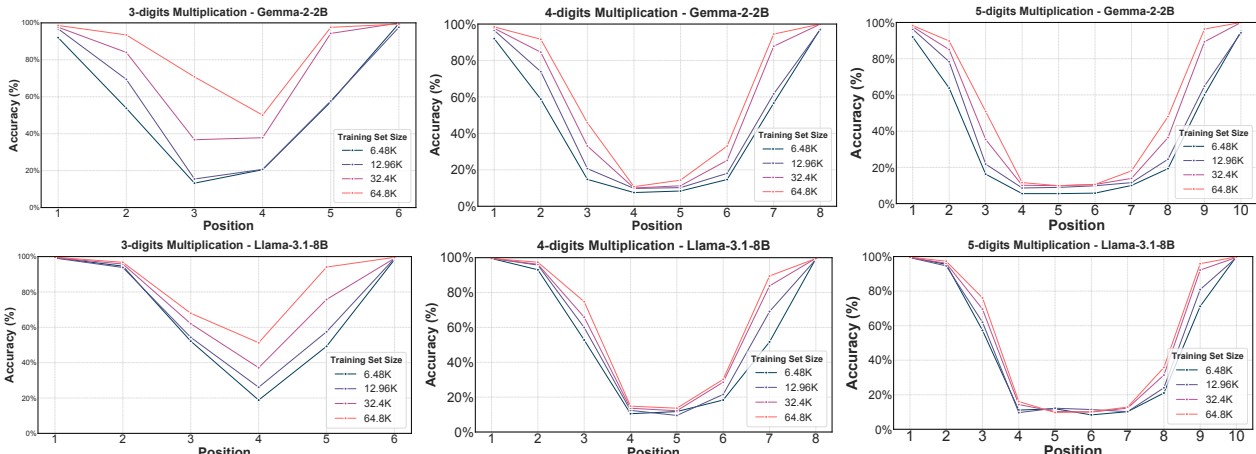

Figure 2: Position-level Accuracy from Gemma-2-2B and Llama-3.1-8B.

conditions but rather reflects a fundamental aspect of how transformer models process sequential arithmetic operations.

The emergence of this U-shaped accuracy distribution provides compelling evidence that *the difficulty in learning multiplication is concentrated in the middle positions rather than at the beginning or end.* This finding has profound implications for our understanding of arithmetic learning in large language models. The high accuracy at terminal positions suggests that models can effectively learn to identify the correct starting and ending digits through pattern recognition, possibly by memorizing frequent digit combinations or leveraging positional embeddings. Meanwhile, the poor performance in middle positions indicates that models struggle with the complex interdependencies and carry operations that occur during the intermediate steps of multi-digit multiplication.

Furthermore, this pattern intensifies with increasing digit complexity, as evidenced by the more pronounced accuracy drops in 5-digit multiplication compared to 3-digit tasks. The robustness of this phenomenon across different training set sizes also suggests that simply increasing the volume of training data may not be sufficient to overcome the inherent difficulty of middle-position calculations, pointing toward the need for more sophisticated training strategies or architectural modifications specifically designed to handle sequential arithmetic dependencies.

### 3.3 Why Subgroup Quality $Q(s)$ Explains the U-Shaped Accuracy

**What $Q(s)$ measures.** For an output digit position $C_j$, a subgroup $s = ((A, B), C_j)$ keeps only a subset of input digits from $A$ and $B$ and masks the rest to zero. The *subgroup quality $Q(s)$* is the accuracy of predicting $C_j$ using this masked product (Alg. 1). Intuitively, $Q(s)$ quantifies how good a *token-limited shortcut* is for that position. *Example.* If we keep only $(A_2, B_2)$ and zero all other digits, $Q(((A_2, B_2), C_1))$ asks: "How often does the units digit computed from those two digits alone match the true units digit?"—a concrete, measurable notion of shortcut strength.

**The token-budget tree (Fig. 3).** All subgroups that use exactly $k$ tokens ($|A|+|B| = k$) form level $k$ of a *token-budget tree.* Parents at level $k+1$ strictly contain a child at level $k$ (*Token Inclusion*). Under our masking protocol, adding observed digits cannot increase ambiguity for $C_j$, giving the monotonicity

$$Q(s_{\text{parent}}) \ \geq \ Q(s_{\text{child}}) \qquad \text{whenever } s_{\text{child}} \subset s_{\text{parent}} \text{ in tokens.} \qquad (4)$$

The root (all digits kept) attains $Q = 1$, while very small-token leaves approach chance quality ($\approx 0.1$ on uniform 10-way digits). *How to read Fig. 3.* Left-to-right within a layer, nodes differ in *which* digits they observe; top-to-bottom, layers differ in *how many* digits they observe, so moving upward generally (weakly) increases $Q$ by equation 4.

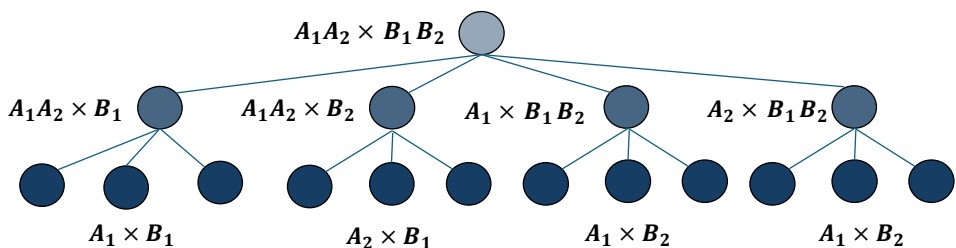

Figure 3: **Tree Structure for 2-digits multiplication.** Given an output position $\mathbb{C}$, subgroups can be organized into a hierarchical tree structure. Each layer represents the number of tokens used by the corresponding subgroup. We do not draw lowest layer (e.g., $A_1$) in this figure as its quality equals to 0.

**Why edges are easy with few tokens.** The U-shape comes from the *existence* (or lack) of high-quality, *low-token* subgroups across positions:

- **Least significant digit ($C_1$).** For $A_1 A_2 \times B_1 B_2$, the units digit depends only on $(A_2, B_2) \,(\mathrm{mod}\ 10)$. Thus $s = ((A_2, B_2), C_1)$ (two tokens) achieves $Q = 1$ even after masking higher digits, because those contribute multiples of 10.

- **Most significant digit ($C_4$).** With fixed-width inputs, order-of-magnitude constraints and limited carry patterns mean that leading digits (e.g., $A_1, B_1$) already give a high-quality predictor for $C_4$ with small $k$.

- **Middle digits ($C_2, C_3$).** These integrate many cross-terms and carries. Any low-$k$ subgroup necessarily omits essential contributors, so $Q(s)$ is low until more tokens are included.

*Illustration.* In $27 \times 38$, the units digit is $7 \times 8 = 56 \Rightarrow 6$—unchanged by masking the tens digits. For the leading digit, $93 \times 47 \approx (9 \times 4) \times 10^2 = 36 \times 10^2$, so the most significant digit is tightly constrained even if not always exactly determined. By contrast, for $47 \times 56 = 2632$, the second digit ($C_2 = 3$) depends on multiple cross-terms and carry; seeing only $(A_2, B_1) = (7, 5)$ leaves $C_2$ ambiguous.

**Aggregating by budget & matching Fig. 4.** Define the best quality achievable at budget $k$ for position $C_j$:

$$Q_k(C_j) \;=\; \max_{|A|+|B|=k} Q\big(((A, B), C_j)\big).$$

For small $k$, $Q_k(C_1)$ and $Q_k(C_m)$ (MSB) are high, while $Q_k(C_j)$ is low for middle $j$; hence the vector $(Q_k(C_1), \ldots, Q_k(C_m))$ is *U-shaped.* As $k$ grows, monotonicity in equation 4 lifts all positions, but the middle digits catch up last. *Link to the plot.* In Fig. 4, the "2-digit-low/middle/high" series visualize exactly this: higher token budgets ("high") raise the entire curve while preserving the edge advantage at small budgets; the "3/4/5-digit-low" curves show the same early U-shape for longer inputs when the budget is tight.

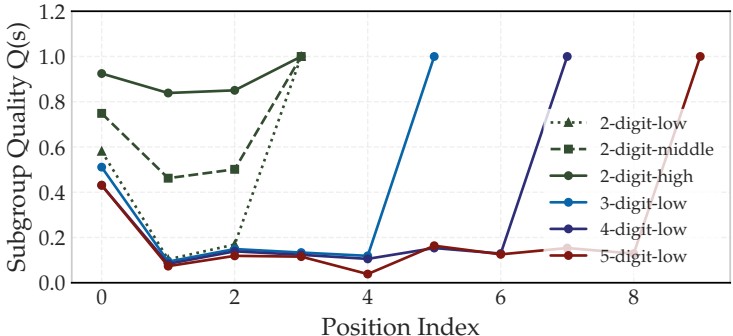

Figure 4: **Position-level Subgroup Quality.** The low-to-high order reflects the hierarchy in the tree structure. Similar trends in $3 - 5$ digits with 2-digits (see Appendix B).

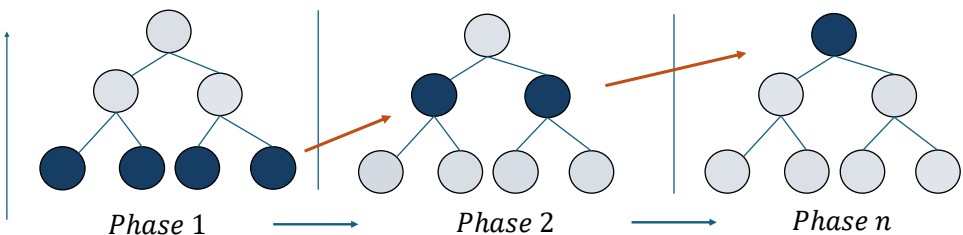

Figure 5: **Searching Program inside LMs.** Different phases refer to different stage of LMs fitting.

**Learning dynamics as tree search (Fig. 5).** Next-token training tends to adopt the *fewest-token, highest-Q* shortcuts first. Phase 1 in Fig. 5 highlights dark leaves: the model quickly fits $C_1$ (units) and often $C_m$ (leading) using low-$k$ subgroups. As residual errors persist in middle positions, gradients drive the model upward to larger-token subgroups (Phase 2), and eventually toward near-root programs (Phase $n$). This *climb up the token-budget tree* explains both the early *U-shape* and its later *flattening* with more training. *Operational cue.* You should see Fig. 4 flatten over training checkpoints: edge positions plateau early, middle positions rise later as the model "unlocks" higher layers of token budget tree.

**Concrete 2-digit example.** For $A_1A_2 \times B_1B_2$:

1. $C_1$: $((A_2, B_2), C_1)$ with $k = 2$ yields $Q = 1 \Rightarrow$ easy early gains.

2. $C_2$: any $k = 2$ subgroup (e.g., $((A_2, B_1), C_2)$) misses carry/cross-terms $\Rightarrow$ low $Q$, needs higher $k$.

3. $C_4$: $((A_1, B_1), C_4)$ already constrains magnitude $\Rightarrow$ relatively high $Q$ at small $k$.

*Numbers.* In $48 \times 39 = 1872$, $C_1{=}2$ is fixed by $(8, 9)$ alone, but $C_2{=}7$ depends on $8{\times}3$, $4{\times}9$, and carry from $8{\times}9$, so observing only one of these pairs cannot nail $C_2$.

**Predictions and diagnostics.** (i) If we cap usable tokens (e.g., stronger masking, context truncation), edge digits degrade least. (ii) As the cap relaxes (bigger context or a reveal-curriculum), the U-shape flattens *from the middle outward.* (iii) Data that amplifies mid-digit carries (hard negatives) accelerates the climb for middle positions. *How to use this.* To test the theory, gradually increase which digits are unmasked during training: you should see $C_1/C_m$ saturate early and $C_2/C_3$ improve only when their requisite digits are revealed, matching the phase progression in Fig. 5.

**Takeaway.** $Q(s)$ is a computable lens on shortcut viability under token constraints. Figures 3, 4, and 5 together show the mechanism: models first exploit *low-$k$ / high-$Q$* edge subgroups, then progressively integrate more digits to master the middle—yielding the observed U-shaped accuracy curve and its evolution over training. *Plain language.* Edges are easy because there exist "cheap but good" rules for them; the middle is hard because it needs "expensive" rules that combine more digits, so the model learns those later.

## 4 From Subgroup Quality to Subgroup Entropy

Although subgroup quality $Q(s)$ is a useful measure for revealing a language model's tendency to use minimal tokens, it has limitations when compared to Kolmogorov complexity in a UTM. Specifically:

- The $O(2^n \times n)$ complexity of calculating $Q(s)$ makes it computationally costly for long sequences.
- It is difficult to compare arithmetic across tasks, and the connection between subgroup quality and error estimation is not clear.[1]

Therefore, we propose subgroup entropy as a complementary measure. It is computationally efficient, and we plan to investigate its error estimation capabilities across tasks.

---

[1]In SI (Solomonoff, 1964a), Solomonoff shows that the MSE of $M(x)$ is upper-bounded by $K(\mu)ln2$ upon convergence.

Table 1: Label space statistics with different rule perturbations. $H(\mathcal{L})$ represents the entropy of the label space, and $|\mathcal{L}|$ is the size of the label space. $\{C_j\}_{i=1}^n$ represents all positions in output digits.

| | | $C_1$ | $C_2$ | $C_3$ | $C_4$ | $C_5$ | $\{C_i\}_{i=1}^n$ | |
|---|---|---|---|---|---|---|---|---|
| Task | Format | $H(\mathcal{L})$ | $H(\mathcal{L})$ | $H(\mathcal{L})$ | $H(\mathcal{L})$ | $H(\mathcal{L})$ | $|\mathcal{L}|$ | $H(\mathcal{L})$ |
| $f(a,b) = a + b$ | $A_1A_2 + B_1B_2 = C_1C_2C_3$ | 0.9710 | 3.3215 | 3.3219 | – | – | 179 | 7.2130 |
| $f(a,b) = a + b + 1$ | $A_1A_2 + B_1B_2 = C_1C_2C_3$ | 0.9649 | 3.3215 | 3.3219 | – | – | 179 | 7.2130 |
| $f(a,b) = a + b + 15$ | $A_1A_2 + B_1B_2 = C_1C_2C_3$ | 0.9280 | 3.3214 | 3.3219 | – | – | 179 | 7.2130 |
| $f(a,b) = a + b + 115$ | $A_1A_2 + B_1B_2 = C_1C_2C_3$ | 0.9280 | 3.3214 | 3.3219 | – | – | 179 | 7.2130 |
| $f(a,b) = (a + b) \bmod 100$ | $A_1A_2 + B_1B_2 = C_1C_2$ | 3.3214 | 3.3219 | – | – | – | 100 | 6.6432 |
| $f(a,b) = (a + b) \bmod 50$ | $A_1A_2 + B_1B_2 = C_1C_2$ | 2.3217 | 3.3219 | – | – | – | 50 | 5.6436 |
| $f(a,b) = (a + b) \bmod 10$ | $A_1A_2 + B_1B_2 = C_1$ | 3.3219 | – | – | – | – | 10 | 3.3219 |
| $f(a,b) = a \times b$ | $A_1A_2 \times B_1B_2 = C_1C_2C_3C_4$ | 2.8979 | 3.3215 | 3.3160 | 3.0340 | – | 2621 | 11.1172 |
| $f(a,b) = a \times b \times 2$ | $A_1A_2 \times B_1B_2 = C_1C_2C_3C_4C_5$ | 0.6873 | 3.2173 | 3.3215 | 3.2964 | 2.2227 | 2621 | 11.1172 |
| $f(a,b) = a \times b \times 4$ | $A_1A_2 \times B_1B_2 = C_1C_2C_3C_4C_5$ | 1.6030 | 3.3020 | 3.3204 | 3.2234 | 2.2227 | 2621 | 11.1172 |
| $f(a,b) = a \times b \times 8$ | $A_1A_2 \times B_1B_2 = C_1C_2C_3C_4C_5$ | 2.5811 | 3.3202 | 3.3151 | 3.2235 | 2.2227 | 2621 | 11.1172 |
| $f(a,b) = (a \times b) \bmod 100$ | $A_1A_2 \times B_1B_2 = C_1C_2$ | 3.3160 | 3.0340 | – | – | – | 100 | 6.2912 |
| $f(a,b) = (a \times b) \bmod 50$ | $A_1A_2 \times B_1B_2 = C_1C_2$ | 2.3210 | 3.0340 | – | – | – | 50 | 5.3494 |
| $f(a,b) = (a \times b) \bmod 10$ | $A_1A_2 \times B_1B_2 = C_1$ | 3.0340 | – | – | – | – | 10 | 3.0340 |

## 4.1 Subgroup Entropy as a "Cheap" Error Estimation

**Settings.** We first deliberately perturb the underlying data generation function $f$ to observe the correlation between subgroup entropy $H'(s)^2$ and accuracy. We consider addition $f(a,b) = a + b$ and multiplication $f(a,b) = a \times b$ as our baselines. For addition, the perturbation is defined as $f(a,b) = a + b + \Delta c$, where $\Delta c \in \{1, 15, 115\}$ corresponds to offsets at different positions/magnitudes. For multiplication, the perturbation is defined as $f(a,b) = a \times b \times \lambda$ with $\lambda \in \{2, 4, 8\}$ for analogous reasons. Additionally, we incorporate modular addition and multiplication as further perturbations. Table 1 summarizes the label-space statistics after applying perturbations. We then fine-tune Gemma-2-2B (Team et al., 2024) and Llama-3.1-8B (Dubey et al., 2024) on data generated by these perturbation functions to analyze how subgroup entropy $H'(s)$ evolves.

**Entropy–Accuracy Analysis.** The results in Table 2 show that both Gemma-2-2B and Llama-3.1-8B produce consistent outcomes across two rule perturbation methods and three setups. Interestingly, LLMs handle arithmetic like $13 \times 10 = 520$ similarly to $13 \times 10 = 130$ when the subgroup entropy $H'(s)$ remains fixed. This demonstrates that tasks with similar $H'(s)$ share comparable error bounds. For modular addition and multiplication with varying modulus values, reducing the entropy size improves performance in both cases. These findings suggest that arithmetic tasks with lower $H'(s)$ in the subgroup space are easier to learn and result in fewer errors. This phenomenon confirms that subgroup entropy is a better measure for estimating errors in arithmetic learning tasks.

Table 2: Test Accuracy difference $\Delta$ on perturbed operations.

| Data Generation Function | $H'(s)$ | Gemma-2-2B | Llama-3.1-8B |
|---|---|---|---|
| $f(a,b) = a + b$ | 7.21 | – | – |
| $f(a,b) = a + b + 1$ | 7.21 | −0.1% | −0.1% |
| $f(a,b) = a + b + 15$ | 7.21 | −0.9% | +0.1% |
| $f(a,b) = a + b + 115$ | 7.21 | −1.4% | +0.7% |
| $f(a,b) = (a + b) \bmod 100$ | 6.64 | +10.1% | +3.7% |
| $f(a,b) = (a + b) \bmod 50$ | 5.64 | +13.1% | +6.7% |
| $f(a,b) = (a + b) \bmod 10$ | 3.32 | +26.1% | +13.7% |
| $f(a,b) = a \times b$ | 11.12 | – | – |
| $f(a,b) = a \times b \times 2$ | 11.12 | −1.1% | −2.7% |
| $f(a,b) = a \times b \times 4$ | 11.12 | −1.7% | +0.7% |
| $f(a,b) = a \times b \times 8$ | 11.12 | +0.2% | −3.7% |
| $f(a,b) = (a \times b) \bmod 100$ | 6.29 | +7.1% | +3.8% |
| $f(a,b) = (a \times b) \bmod 50$ | 5.35 | +12.1% | +5.3% |
| $f(a,b) = (a \times b) \bmod 10$ | 3.03 | +18.9% | +10.7% |

**What Table 1 reveals (entropy shifts under perturbations).** Three consistent patterns emerge. (i) *Additive translations preserve total label entropy while redistributing it across positions.* For $f(a,b) = a + b$ and $a + b + \Delta c$, the total entropy remains $H(\mathcal{L}) = 7.2130$, but the lowest digit becomes slightly more predictable as $\Delta c$ grows (e.g., $C_1$ drops from 0.9710 to 0.9280), whereas $C_2/C_3$ stay near 3.32; this reflects a carry pattern shift localized to early positions. (ii)

---

$^2 H'(s)$ represents $H(s)$ summed over all output positions $\mathbb{C}$.

*Multiplicative scalings largely conserve total entropy while shifting where entropy lives.* For $a \times b$ vs. $a \times b \times \lambda$, the total stays $H(\mathcal{L}) = 11.1172$, but entropy mass migrates toward higher positions (a new $C_5$ appears with $\approx 2.22$ bits) and the units digit entropy rises monotonically with $\lambda$ ($C_1$: $0.69 \rightarrow 1.60 \rightarrow 2.58$ for $\lambda = 2, 4, 8$), capturing how scaling changes residue classes and carry dispersion across digits. (iii) *Modular reductions compress the label space and thus the total entropy roughly to* $\log_2 |\mathcal{L}|$. For addition, (mod $100, 50, 10$) yields totals $6.64, 5.64, 3.32$ (very close to $\log_2 100, \log_2 50, \log_2 10$), consistent with nearly uniform residues; for multiplication the totals are slightly smaller ($6.29, 5.35, 3.03$) due to non-uniform multiplicative residue distributions (e.g., last-digit biases), explaining why modular tasks empirically become easier as $|\mathcal{L}|$ shrinks.

## 4.2 Extend Study on CoT: A Symbolic View

Building on subgroup entropy's effectiveness in error estimation for arithmetic tasks, we extend our experiments from simple arithmetic to a chain-of-thought (CoT) setting (Wei et al., 2023). CoT involves a multi-step reasoning to solve a single problem. For example, in arithmetic learning, solving a multiplication problem using CoT means breaking it down into smaller steps to arrive at the correct answer. Recent studies have shown that RL-based CoT improves reasoning abilities in advanced LMs such as OpenAI $o1$ (OpenAI, 2024) and DeepSeek-$r1$ (DeepSeek-AI et al., 2025).

**Settings.**  We still take multiplication as the baseline. While multiplication is mathematically well-defined, there are multiple methods to perform the calculation. Historically, four methods stand out as the most representative: *Standard Multiplication, Repetitive Addition, the Lattice Method, and Egyptian Multiplication* (details in Appendix C.1). These methods correspond to four distinct CoT paths. We fine-tune Gemma-2-2B and Llama-3-8B on data from these CoT paths to observe how $H'(s)$ evolves.

Table 3: **Performance comparison of different CoT paths.** Different CoT split the original tasks into different CoT steps.

| CoT Type | # CoT Step | Avg. $H'(s)$ | Avg. $Q(s)$ | Accuracy (%) |
|---|---|---|---|---|
| Original | 1 | 11.1172 | 9.87 | 73.12 |
| Standard Multiplication | 3 | 7.8431 | 7.12 | 86.32 |
| Repetitive Addition | 3 | 5.8313 | 6.03 | 91.31 |
| Lattice Method | 5 | 5.1130 | 5.41 | 94.41 |
| Egyptian Multiplication | 8 | **3.2130** | **4.02** | **98.31** |

**Results.**  The results of the performance comparison between four different CoT methods are shown in Table 3. Using no intermediate steps (Original), the task yields an average entropy of 11.12 and a test set accuracy of 73.12%. This serves as a baseline to compare the effectiveness of CoT methods. Two key observations can be drawn:

- *Splitting into More Steps Improves Performance*: Decomposing a task into multiple steps, as demonstrated by the Egyptian Multiplication method, leads to improved performance compared to the baseline. This likely occurs because splitting the task results in lower average entropy $H'(s)$ per sub-task.

- *Strategic CoT Splitting is Crucial*: Comparing *standard multiplication* to *repetitive addition* reveals that even when a task is split into a fixed number of steps, the choice of splitting strategy significantly impacts performance. Choosing a split that yields lower average entropy $H'(s)$ per sub-task results in superior overall performance.

The subgroup quality $H'(s)$ proves to be effective at estimating and correlating CoT errors. By focusing on the entropy of individual steps in the reasoning process, this approach offers a more detailed and effective way to evaluate and optimize performance in complex reasoning tasks. These findings demonstrate the potential of the subgroup induction framework to advance reasoning-intensive math tasks.

### 4.3 Discussions

Overall, we propose two measures to assess subgroup complexity: subgroup quality $Q(s)$ and subgroup entropy $H(s)$. These measures complement each other. Subgroup quality is best suited for understanding learning mechanisms, revealing how LMs initially try to use fewest tokens and gradually incorporate more tokens if needed. However, it is computationally expensive and less effective for error estimation. On the other hand, subgroup entropy is less effective at capturing learning dynamics but excels in error estimation across tasks, including both simple arithmetic and CoT settings. Together, these two measures provide a robust framework for evaluating subgroup complexity.

## 5 Related Work

**Understanding Arithmetic Learning in Transformer**   Research on understanding arithmetic primarily in previous work is to identify *causal correlations* between model components and outputs. Stolfo et al. (2023) identify key attention layers responsible for arithmetic learning using causal mediation analysis (CMA), a weight perturbation method that observes changes in output. Similarly, Hanna et al. (2023) and Wu et al. (2024) explore causal abstraction concepts at different model scales, specifically 0.1B and 7B parameters, respectively. More recently, Zhang et al. (2024) isolate attention heads and fine-tune them for improved performance at a lower cost. While these studies have made progress in understanding how LMs perform arithmetic at a component level, our research provide another perspective from algorithm learning theory.

**Large Language Models and Solomonoff Induction**   Solomonoff induction is widely regarded as one of the most powerful predictors in the field of universal prediction, providing a theoretical foundation for optimal Bayesian inference in algorithmic probability (Solomonoff, 1964a;b). Meanwhile, large language models have sparked discussions about whether they approximate universal Turing machines (Chen et al., 2018; Lu & Lu, 2020; Stogin et al., 2022; Mali et al., 2023) and whether their learning behavior resembles Bayesian updating (Ortega et al., 2019; Arora et al., 2024). And some previous work also discuss the relationship between transformer's behavior and Solomonoff induction (Young & Witbrock, 2024; Grau-Moya et al., 2024). These debates have inspired us to develop a practical framework for understanding the mechanisms underlying language models.

**Arithmetic Reasoning in LLMs**   Mathematical reasoning has been a longstanding area of research in natural language processing (NLP) (Kushman et al., 2014; Huang et al., 2016; Wang et al., 2017; Thawani et al., 2021; Sundaram et al., 2022; Guo et al., 2024). However, LLMs still face challenges with basic calculations and remain vulnerable to adversarial examples or perturbations, where minor changes in problems can result in incorrect answers (Zhou et al., 2023; Xie et al., 2024). Several previous efforts have aimed to improve arithmetic learning in LLMs. Lee et al. (2023) trained a 10.6M NanoGPT (Karpathy, 2022) model to learn arithmetic by carefully curating the data format, explicitly expanding each step using a method termed Scratchpad, which achieved remarkable performance compared to GPT-2 XL (Radford et al., 2019). Yang et al. (2023) fine-tuned MathGLM with a sufficient training dataset, demonstrating its capability to solve 5-digit multiplication. Deng et al. (2023; 2024) further advanced this field by internalizing the CoT process, hiding detailed steps in a scheduled manner, enabling GPT-2 small to solve 9-digit multiplication after multiple training runs.

## 6 Conclusions

In conclusion, our work introduces the novel framework of subgroup induction, offering a pragmatic adaptation of Solomonoff induction tailored to arithmetic learning in language models. By leveraging subgroup quality and entropy as complementary measures, we uncover a U-shaped learning pattern that reflects Occam's Razor principles in LMs. This innovative approach bridges theoretical universal prediction with modern neural architectures, providing robust insights into arithmetic learning mechanisms. Our findings emphasize the practicality of our framework in addressing complexity and advancing reasoning research, establishing a foundational step forward in understanding and improving the capabilities of LMs in arithmetic tasks.

## Acknowledgements

We thank the anonymous reviewers for their thoughtful feedback and constructive suggestions, which helped improve this work. This project is supported in part by the Rice Ken Kennedy Institute Research Award #1081025. We also thank members of the CHILI Lab for helpful discussions and support throughout the project.

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

## A  Experiment Setup

In this section, we detail the experiment setup used in the main body of our work. Our experiments are designed to investigate the capabilities of language models in learning and performing arithmetic tasks, with a focus on understanding the underlying mechanisms of arithmetic reasoning. By carefully selecting the domain, models, training methodologies, and datasets, we aim to provide a comprehensive analysis of how language models can be adapted to handle arithmetic operations effectively. Below, we describe each component of our setup in detail.

**Domain.** We select addition and multiplication as the fundamental operations for our experiments, following previous work (Lee et al., 2023; Deng et al., 2023; 2024). We do not consider division or subtraction, as subtraction can be viewed as a similar operation to addition, and division introduces floating-point numbers, which involve more complex formatting issues in this task. Therefore, we primarily focus on multiplication and aim to understand it in depth. However, by definition, our subgroup induction framework is suitable for all arithmetic tasks.

**Model.** To investigate arithmetic learning at scale, we selected two open-source LLMs, Gemma-2-2B (Team et al., 2024) and Llama-3.1-8B (Dubey et al., 2024). Both models are top performers in their respective categories and excel at language-related tasks. We did not choose GPT-4o (OpenAI et al., 2024) or other proprietary LLMs due to concerns that they may internally integrate function calling (e.g., invoking APIs or executing Python programs), which could affect the experimental setup.

**Training Details.** We use LoRA Hu et al. (2021) to optimize memory usage during the training of language models. The LoRA rank is set to 8, and we employ rank-stabilized LoRA in our setup. We set the learning rate to 3e-4, the number of epochs to 12, and the batch size to 16. Additionally, we use gradient checkpointing with a step size of 4 to save memory.

**Data.** For n-digit arithmetic, we consider the optimal dataset size. Specifically, for 2-digit multiplication, we have 8,100 data points in total, and for 3-digit multiplication, we have 810,000. We split the dataset into train, development, and test sets with a ratio of 8:1:1. We train the language models on the training set, select the best checkpoints based on validation loss, and use them to evaluate the test set accuracy.

**Conventional Data Format.** We directly train the model to predict the output (e.g., 130) given the input operands and the operator (e.g., $13 \times 10$). e add one space between each digit to ensure tokens are split into individual digits We do not use chain-of-thought (CoT) (Wei et al., 2023) or other prompting strategies to enforce the model to focus on arithmetic learning in main experiment body. But we will discuss how subgroup entropy could be a suitable metrics for quantifying CoT path.

## B Subgroup Quality for High-digits Multiplication

### B.1 Criteria for Low/Middle/High Node Selection

We define the low node as subgroups with two token positions, such as $A_1 B_1$, which are located closest to the leaf nodes in the tree structure. The middle node is selected as subgroups with $\lceil n \rceil$ token positions in $n-$digit arithmetic. Subgroups with $n - 1$ token positions are classified as high nodes, which are positioned closer to the root node. In summary, the middle node resides at the intermediate level of the tree structure, the low node is near the leaf level, and the high node is near the root level.

### B.2 Subgroup Quality for 3/4/5-digits Multiplication

As shown in Figure 6, we can also observe a similar trend in these multiplications with the language model (LM) learning arithmetic, characterized by U-shaped learning. In high-digit multiplications, the subgroup quality for the middle digits remains very low for lower nodes, while higher nodes exhibit better quality. This phenomenon strongly reveals the underlying mechanism of language models' learning dynamics: they tend to use more and more tokens in predictions to reduce the loss during gradient descent. We believe this is an important finding that may be helpful for future research.

## C Are Large Language Models Implicit Calculators?

In this section, we explore whether LLMs utilize partial products to enhance their arithmetic calculation capabilities, particularly in the context of *multiplication*. It is important to note that while multiplication is well-defined mathematically, the process of multiplication calculation is not limited to traditional methods

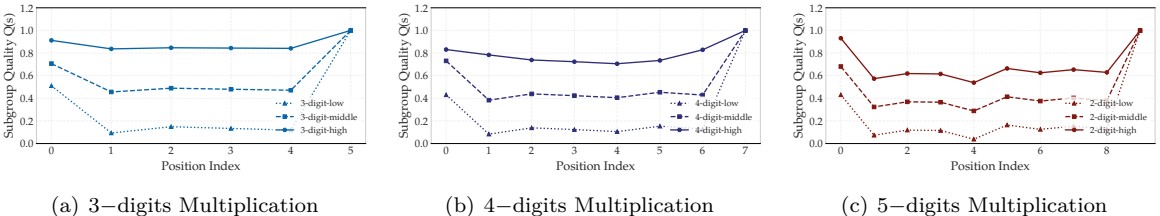

Figure 6: Subgroup quality for 3/4/5-digits multiplication given different node.

defined in textbook. Thus, examining only one calculation method presents a flawed experimental design that is vulnerable to exploitation. We selected four calculation methods that are representative to cover the major approaches to multiplication.

### C.1 Historical and Modern Multiplication

In terms of multiplication, four different calculation methods are most representative from history to now: Standard Multiplication, Repetitive Addition, Lattice Method, and Egyptian Multiplication.

**M1: Standard Multiplication** In standard multiplication, we multiply each digit of one number by each digit of the other number, and then sum the results appropriately:

$$12 \times 34 = 12 \times (30 + 4) = 12 \times 30 + 12 \times 4$$
$$= 360 + 48 = 408$$

**M2: Repetitive Addition** Multiplication can be interpreted as repeated addition. For $12 \times 34$, we add 12 thirty-four times:

$$12 \times 34 = 12 + 12 + 12 + \cdots + 12 \quad (34 \text{ times})$$
$$= 408$$

**M3: Lattice Method** In the lattice method (or grid method), we place the numbers along the top and side of a grid, perform single-digit multiplications, and then sum along the diagonals:

$$12 \times 34 = 10 \times 30 = 300$$
$$10 \times 4 = 40$$
$$2 \times 30 = 60$$
$$2 \times 4 = 8$$

Summing the results: $300 + 40 + 60 + 8 = 408$

**M4: Egyptian Multiplication** Egyptian multiplication computes the product by doubling the multiplicand and adding the results corresponding to the powers of two that sum to the multiplier. For $12 \times 34$:

$$12 \times 34 = 12 \times 1 = 12$$
$$12 \times 2 = \mathbf{24}$$
$$12 \times 4 = 48$$
$$12 \times 8 = 96$$
$$12 \times 16 = 192$$
$$12 \times 32 = \mathbf{384}$$

Summing the selected results: $24 + 384 = 408$

Since $34 = 2 + 32$, we select the results for $12 \times 16$ and $12 \times 8$, and summing these gives the final product.

Table 4: Inductive and deductive accuracy difference $\Delta$.

| | Gemma-2-2B | | | | Llama-3.1-8B | | | |
|---|---|---|---|---|---|---|---|---|
| | Standard | Lattice | Repetitive | Egyptian | Standard | Lattice | Repetitive | Egyptian |
| Task $\rightarrow$ Partial P. | +4.1% | +6.8% | −29.0% | +3.6% | +40.6% | +40.8% | −59.0% | +29.6% |
| Partial P. $\rightarrow$ Task | −6.1% | −10.7% | −20.3% | −9.6% | −3.7% | −0.2% | −0.9% | −2.7% |

## C.2 Examining Partial Product in Arithmetic Learning

To investigate whether LLMs generate partial products during arithmetic learning, we employ a set of diagnostic tasks as an approach to trace. We fine-tune Gemma-2-2B and Llama-3.1-8B on two-digit multiplication, observing changes in accuracy on diagnostic sets before and after fine-tuning (Task $\rightarrow$ Partial Products). Subsequently, we fine-tune the LLMs on these diagnostic sets and examine how their accuracy on the multiplication task changes (Partial Products $\rightarrow$ Task).

Table 5: Diagnostic sets with four calculation methods.

| Method | Diagnostic Sets |
|---|---|
| Standard Multiplication | $\mathcal{P}_{\text{std}} = \{A_1 \times B_1B_2, A_2 \times B_1B_2, B_1 \times A_1A_2, B_2 \times A_1A_2\}$ |
| Repetitive Addition | $\mathcal{P}_{\text{ra}} = \{\sum_{i=1}^{B_1B_2} A_1A_2, \sum_{i=1}^{A_1A_2} B_1B_2\}$ |
| Lattice Method | $\mathcal{P}_{\text{lattice}} = \{A_10 \times B_10, A_10 \times B_2, A_2 \times B_10, A_2 \times B_2\}$ |
| Egyptian Multiplication | $\mathcal{P}_{\text{egyptian}} = \{2^k \times A_1A_2 \mid k \in 0, 1, \ldots, \lfloor \log_2(B_1B_2) \rfloor\}$ |

We probe language models' partial product in four different directions. As shown in Table 5, for a task formatting like $A_1A_2 \times B_1B_2 = C_1C_2C_3C_4$, we would generate diagnostic test for each algorithm.

**Accuracy on Identifying Partial Products** According to the results in Figure 7, we found that standard multiplication, the lattice method, and the Egyptian method significantly improved in identifying partial products after fine-tuning, with gains of +17.45%, +18.35%, and +10.45%, respectively. However, for repetitive addition tasks, LLMs failed to identify partial products, achieving an accuracy of only about 5% after fine-tuning.

**A Deeper Look into Calculations** Do the results showing increased accuracy across three paths really imply that partial products are used in arithmetic learning? We have two arguments against this interpretation. First, if LLMs genuinely use partial products to learn arithmetic, it is likely that they only use one calculation path at a time. Thus, the simultaneous improvement across three paths (standard, lattice, and Egyptian) is unusual. Second, if LLMs employ a specific path to compute partial products, this process should be demonstrated as bidirectional. Specifically, LLMs fine-tuned on a task should be able to identify partial products (inductive), and conversely, mastering partial products should enhance task learning (deductive). However, we currently have results for only one direction, lacking evidence for the other. Therefore, we extend our experiments to another direction.

**Accuracy on Identifying Tasks** We fine-tune two LLMs on diagnostic sets and present the results of identifying tasks before and after fine-tuning in Table 4. Our findings reveal that, fine-tuning specifically on partial products does not enhance task learning. Instead, it results in a performance drop across all four calculation paths for both models. This indicates that pre-learning partial products does not aid in arithmetic learning. The improved ability to recognize partial products appears to stem from the symbol learning process (note that the standard partial product $A_1 \times B_1B_2$ is a sub-portion of $A_1A_2 \times B_1B_2$, similar to lattice and Egyptian methods) rather than being an intrinsic computational method used by the models.

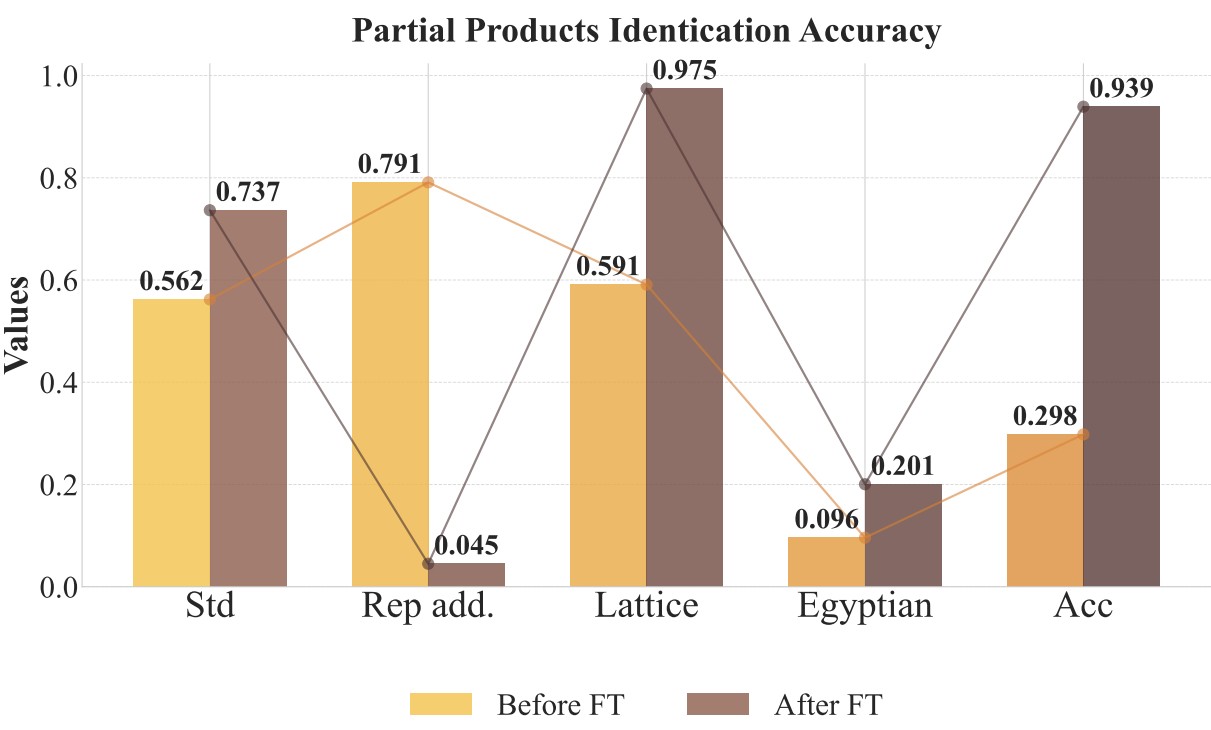

Figure 7: Partial products identification accuracy before and after fine-tuning on tasks. Scores are reported on average of Gemma-2-2B and Llama-3.1-8B.

