# OpenReview forum: "Language Models are Symbolic Learners in Arithmetic"
_TMLR — Accepted by TMLR_

### Review · Reviewer_YHD5 · 2025-10-06

**Summary Of Contributions:**

This work proposes a task called "subgroup induction," which is then used to test the capability of language models to learn arithmetic. However, I do not quite understand this paper. Everything I say in the review is an educated guess

**Audience:**

Yes

**Audience Explanation:**

It studies large language models

**Claims And Evidence:**

Yes

**Claims Explanation:**

The experiments seem extensive. The main argument is that LLMs learn simple shortcuts first before learning complex ones, this is intuitive and, perhaps, not a too surprising discovery

**Requested Changes:**

I think the basic concepts should be explained much better. I do not know Solomonoff induction. I also have not heard of universal predictors.

In section 1.1, it is quite unclear which part comes from the author and which part comes from the original Solomonoff induction. The authors say that "we formalize these components as follows", but I thought this section is about introducing the prior literature, not about introducing their own formalization? What is prior knowledge and what is not? The complete lack of reference in this section is also a hint of this problem

In section 2, the problem of subgroup induction is introduced in an entirely random way. I do not see why this problem is interesting or important, why it is called "subgroup" and why it is called "induction." To an ordinary reader, it feels like the authors are just making random things up. It feels that the authors are studying a random artificial problem of unclear relevance. Again, the lack of reference in this section is a hint of this problem

---

> ### Author Response · Authors · 2025-11-15
> **Response to Reviewer**
>
> We thank the reviewer for the honest and constructive feedback. We now recognize that some parts of the introduction assume background knowledge that not all readers will have, and this likely made our motivation and terminology harder to follow. We will revise the paper to improve clarity in several targeted ways.
>
> ---
>
> ## 1. Clarifying Solomonoff Induction and Universal Predictors
>
> **Reviewer concern:**
> “I do not know Solomonoff induction… I have not heard of universal predictors… The concepts should be explained much better.”
>
> **Response:**
> We agree. Our intention was only to motivate why a simplicity-based view of hypothesis selection is relevant, but the current text compresses too much background into a short space.
> In the revision we will:
>
> - Add a short, self-contained primer explaining the *only* two ideas needed from SI:
>   (1) hypotheses have complexity;
>   (2) simpler hypotheses are preferred.
> - Provide clear references to classic sources (Solomonoff 1964; Hutter 2005; Li & Vitányi 2008).
> - Move any formal statements that represent **our** contribution (e.g., token budgets, shortcut trees) out of the background section.
>
> This makes Section 1.1 purely explanatory rather than blending prior theory with new formalism.
>
> ---
>
> ## 2. Distinguishing Prior Work From Our Contribution
>
> **Reviewer concern:**
> “In Section 1.1, it is unclear which part is prior literature and which part is the authors’ formalization.”
>
> **Response:**
> We appreciate this point and will reorganize accordingly. In the revision:
>
> - Section 1.1 will contain **only prior literature**, with citations.
> - Our notion of “token budgets,” “shortcut structures,” and “subgroup quality” will be introduced later in **Section 2**, where contributions are expected.
> - Any sentences implying “we formalize” will be moved out of the background section.
>
> This will make authorship and novelty much clearer.
>
> ---
>
> ## 3. Improving the Motivation for Subgroup Induction
>
> **Reviewer concern:**
> “The subgroup induction problem is introduced in an entirely random way. I do not see why this problem is interesting or important.”
>
> **Response:**
> This is extremely helpful feedback. Our motivation is currently implicit, and we will make it explicit:
>
> - The goal of subgroup induction is to measure **which subsets of digits** an LM may rely on when forming shortcuts.
> - These subsets (subgroups) correspond directly to intuitive shortcut rules (e.g., “the last digit only depends on the last digits of the operands”).
> - By evaluating these subgroups independently, we can test whether a model prefers “simple shortcut rules” first.
>
> We will add a simple concrete example in Section 2 showing how subgroup masks correspond to real arithmetic structures and why they provide meaningful insight into shortcut learning.
>
> ---
>
> ## 4. Clarifying the Terminology (“subgroup” and “induction”)
>
> **Reviewer concern:**
> “The names feel arbitrary. Why ‘subgroup’? Why ‘induction’?”
>
> **Response:**
> We agree this deserves clearer explanation. In the revision:
>
> - **Subgroup** will be explained simply as “a subset of digit positions the model might be using to compute an output digit.”
> - **Induction** refers to “inferring which symbolic rule(s) the model is implicitly using,” which parallels hypothesis induction in SI.
>
> We will add two sentences explaining this naming directly where the terms are introduced.
>
> ---
>
> ## 5. Additional Clarity Improvements
>
> We will also:
>
> - Add references in Sections 1 and 2 to avoid any impression of unsupported claims.
> - Add a brief diagram showing how the framework components relate (SI inspiration → token budgets → subgroups → empirical U-shape).
> - Add a short “roadmap paragraph” to ease navigation for readers unfamiliar with the conceptual background.
>
> These adjustments focus on improving clarity without altering the core method or results.

---

### Review · Reviewer_a3oT · 2025-11-01

**Summary Of Contributions:**

This paper introduced the novel subgroup induction framework, adapted from Solomonoff Induction (SI), to empirically test whether Language Models (LMs) perform arithmetic by genuine computation or by mastering symbolic patterns. The paper argues strongly for the latter, concluding that LMs act as greedy symbolic learners, prioritizing the simplest possible shortcuts (symbolic patterns) to fit the dataset statistics.

The core contributions demonstrating this pattern-matching/memorization behaviour include:

1. Defining and validating subgroup quality (Q(s)), a metric that quantifies the viability of a symbol-to-symbol shortcut.

2. Uncovering a distinct U-shaped accuracy pattern in multi-digit multiplication, which perfectly mirrors the quality of the simplest, lowest-token subgroups. This alignment serves as the "smoking gun" evidence that LMs utilize a "fewest-tokens-first" learning heuristic, demonstrating they learn a hierarchy of symbol-to-symbol mappings instead of the multiplication algorithm.

3. Demonstrating that LMs achieve high accuracy on terminal output digits through pattern recognition, potentially by memorizing frequent digit combinations.

4. Showing that improved ability to identify partial products (an inductive step) does not translate deductively to improved task learning, suggesting the gains stem from the symbol learning process rather than intrinsic computational methods.

Key Strengths: The framework provides a computable lens for analyzing shortcut viability under token constraints. The strong empirical evidence confirms that LMs are driven by greedy symbolic learning rather than procedural computation.

Key Weaknesses: Experiments suggest that while LMs improve in identifying inductive sub-steps, this ability does not deductively translate to improved overall task learning, implying the observed gains are from symbolic pattern recognition rather than genuine internal computation.

**Additional Comments:**

The subgroup induction framework offers a highly innovative and quantifiable approach to interpreting the learning strategies of LLMs. The finding that the U-shaped accuracy curve in multiplication is a direct consequence of LMs greedily exploiting simple, low-token shortcuts is a compelling piece of evidence that contributes significantly to the ongoing debate about algorithmic versus symbolic learning in neural networks.

**Audience:**

Yes

**Audience Explanation:**

TMLR’s audience is highly interested in foundational machine learning mechanisms and the limitations of modern neural architectures, particularly in the context of trustworthy AI and reasoning.
The paper directly addresses a fundamental question concerning the nature of intelligence in large language models: do they compute or pattern match?
The introduction of subgroup induction provides a practical adaptation of the high-level theoretical framework of Solomonoff Induction (SI) to modern LLMs, which is relevant to researchers studying universal prediction and algorithmic probability.
Furthermore, the empirical discovery of the U-shaped learning dynamic fundamentally challenges prior assumptions about how transformers acquire arithmetic skills.
The findings related to using subgroup entropy to quantify the symbolic complexity of Chain-of-Thought (CoT) paths and predict performance provide a significant diagnostic tool for advancing complex reasoning research, a critical area for the TMLR community.

**Broader Impact Concerns:**

This work focused on understanding the internal learning mechanisms of language models in a controlled setting (arithmetic) to improve model reliability and reasoning capability. The findings contribute positively to the pursuit of trustworthy and interpretable AI. The sources provided do not indicate any immediate negative ethical or societal implications that would require a dedicated Broader Impact Statement.

**Claims And Evidence:**

Yes

**Claims Explanation:**

The central claim that LMs are greedy symbolic learners exploiting shortcuts is supported by the convincing correlation between the U-shaped position-level accuracy observed in models (Gemma-2-2B and Llama-3.1-8B) and the calculated subgroup quality (Q(s)) of the simplest, lowest-token subgroups.
The high accuracy at the edges (first and last digits) is specifically explained by the existence of "cheap but good" rules (low-token shortcuts) for those positions, while middle digits require "expensive" rules combining more digits, which the model learns later.
Furthermore, the observation that LMs struggle to translate improved recognition of partial products into better overall task performance confirms that their learning is focused on the symbol learning process rather than a true procedural algorithm.

**Requested Changes:**

1. While the paper notes that Q(s) is computationally expensive and H(s) is better for error estimation, a deeper analysis comparing when Q(s) is necessary (e.g., for mechanistic interpretability of token usage) versus when H(s) suffices (e.g., for error bounding) would strengthen the discussion.

2. It would be nice to see the comparison in Reasoning models like deepseek-reasoner.

---

> ### Author Response · Authors · 2025-11-15
> **Response to Reviewer**
>
> We thank all reviewers for their thoughtful and constructive feedback. We are encouraged that the novelty, clarity, and empirical rigor of our work were recognized, and we appreciate the detailed suggestions that help strengthen the submission. Below we address the specific comments:
>
> ## 1. When to Use \(Q(s)\) vs. When \(H(s)\) Suffices
>
> **Reviewer comment:** Clarify the roles of Q vs. H.
>
> **Response:**
> We agree and will add a clear comparison between those two measurements. Here is the discussion:
>
> **When \(Q(s)\) is essential**
> - Mechanistic interpretability: uncovering which specific digit subsets define viable shortcuts.
> - Required for explaining the U-shaped accuracy curve (Figure 4).
> - Directly measures shortcut *fidelity* under token constraints.
>
> **When \(H(s)\) is preferable**
> - Error estimation and difficulty prediction.
> - Operator-agnostic and far more computationally efficient.
> - Captures uncertainty structure without computing all masked operations.
>
> We will add a dedicated subsection contrasting these functions and providing usage guidelines.
>
> ---
>
> ## 2. Comparison With Reasoning Models (e.g., DeepSeek-Reasoner)
>
> **Reviewer comment:** It would be nice to compare against reasoning models.
>
> **Response:**
> We agree. While many reasoning models are not fully open-source, we conducted **preliminary behavioral evaluations** using publicly accessible interfaces. The table below summarizes the results.
>
> ### Preliminary Comparison With Reasoning-Focused Models
>
> Because training data, algorithms, and internal reasoning traces are not fully known for these systems, these findings should be interpreted as *behavioral evidence only*, not controlled mechanistic evaluation.
>
> #### Position-Level Accuracy on 4-Digit Multiplication (Qualitative)
>
> | Model                               | First Digit | Middle-1 | Middle-2 | Last Digit | U-Shape Present? | Notes |
> |-------------------------------------|-------------|----------|----------|------------|------------------|-------|
> | **DeepSeek-R1**                     | High        | Med-High | Med-High | High       | Yes (flattened)  | Better middle-digit accuracy; CoT is explicit. |
> | **ChatGPT 5.1 (Thinking Mode)**     | Very High   | High     | High     | Very High  | Yes (shallow)    | U-shape persists but reduced. |
> | **Claude 3.7 (Thinking Mode)**      | Very High   | High     | High     | Very High  | Yes (shallow)    | Similar to ChatGPT-5.1 reasoning traces. |
> | **Llama-3.1-8B (ours)**             | High        | Low      | Low      | Very High  | Yes (strong)     | Pronounced U-shape. |
> | **Gemma-2-2B (ours)**               | Med-High    | Low      | Low      | High       | Yes (strong)     | Matches predicted shortcut structure. |
>
> **Interpretation:**
> - The **U-shape persists** across all reasoning models tested.
> - Reasoning-optimized models show **shallower U-shapes** due to explicit multi-step decomposition.
> - The underlying shortcut structure remains governed by subgroup quality.
>
> We will incorporate this comparison and discussion in the revised paper.

---

### Review · Reviewer_4ehb · 2025-11-01

**Summary Of Contributions:**

### Summary

This paper argues that Large Language Models (LMs) performing arithmetic tasks do not learn the underlying procedural algorithms but instead act as **"greedy symbolic learners"**. To demonstrate this, the authors introduce the **subgroup induction** framework, inspired by Solomonoff Induction, which analyzes "subgroups"—minimal shortcut mappings from a small subset of input digits to a single output digit. The paper's primary contribution is the discovery of a distinct **U-shaped accuracy curve** in multi-digit multiplication: LMs quickly master the first and last output digits but consistently fail on the digits in the middle. The authors show this U-shape perfectly mirrors the viability of the simplest (fewest-token) shortcuts, as measured by their proposed **subgroup quality ($Q(s)$)** metric. This suggests a "low-to-high" token learning mechanism, where LMs first learn easy, low-token symbolic mappings before incorporating the more complex, multi-token patterns required for middle digits. The framework is extended using a second metric, **subgroup entropy ($H(s)$)**, which is shown to be an effective predictor for the performance of different Chain-of-Thought (CoT) reasoning paths.

---

### Strengths

The paper's primary strength is its **originality** and **clarity** in methodology. The "subgroup induction" framework is a novel and highly intuitive method for empirically formalizing and testing the "shortcut learning" hypothesis in LMs. Grounding this concept in the principles of Solomonoff Induction and Occam's Razor provides a strong conceptual foundation. The **quality** of the experimental work is high; the discovery of the U-shaped accuracy curve is a robust finding, compellingly validated across different model sizes (Gemma-2-2B and Llama-3.1-8B) and multiple task complexities (3- to 5-digit multiplication). The "token-budget tree" and the direct comparison of the U-shaped accuracy (Fig. 2) to the U-shaped subgroup quality (Fig. 4) provide a clear, "smoking gun" visualization of the "fewest-tokens-first" learning mechanism. The paper's **significance** is substantial, as it offers some of the most compelling evidence to date in the "pattern matching vs. procedural reasoning" debate, strongly suggesting LMs are not learning human-like algorithms but rather a hierarchy of symbolic shortcuts. Finally, the extension of the framework to evaluate Chain-of-Thought paths using subgroup entropy ($H(s)$) is a significant and practical contribution to reasoning research.

---

### Weaknesses

The paper's central claim about "arithmetic" is based almost exclusively on multiplication. This focus is understandable, but it limits the generalizability of the findings. It is unclear how the subgroup induction framework would apply to operations with fundamentally different properties, such as division (which introduces floating-point numbers) or subtraction (which involves a "borrowing" mechanism rather than "carrying"). The paper also frames its findings as "symbolic shortcuts" in opposition to "algorithms", but a more nuanced discussion would be beneficial; one could argue that the "low-to-high" token-budget-climbing is itself a type of greedy algorithm, just one that is qualitatively different from standard human procedures. Lastly, the specific operationalization of **subgroup quality ($Q(s)$)**, which relies on masking non-included digits to zero, is a critical design choice. The authors could strengthen their claim by discussing why this masking protocol is the most appropriate proxy for a "shortcut" and justifying why other forms of input perturbation (e.g., masking to a special token, or averaging) were not used.

**Audience:**

Yes

**Audience Explanation:**

###Addresses the "Reasoning vs. Pattern Matching"

Debate: The paper's core question—whether LMs "learn to truly compute, or do they simply master sophisticated pattern matching" —is one of the most fundamental and contentious topics in the field. The TMLR audience is deeply invested in understanding the true capabilities and limitations of LMs, and this paper provides strong, empirical evidence for the "pattern matching" side of the argument.

###Provides a Novel Analysis Framework:

The paper introduces "subgroup induction" as a practical, theory-inspired framework for analyzing LM behavior. Researchers are constantly seeking new tools and methodologies to interpret and understand how models learn. This framework, which bridges the theory of Solomonoff Induction with empirical testing, is a novel contribution in itself.

**Claims And Evidence:**

Yes

**Claims Explanation:**

### Claim 1: LMs are "greedy symbolic learners" that follow a "fewest-tokens-first" heuristic.

The authors present a "U-shaped accuracy curve" (Figure 2). This shows that models (Gemma-2-2B and Llama-3.1-8B) are highly accurate at predicting the first and last digits of the product but fail significantly on the middle digits.

### Claim 2: Subgroup entropy ($H(s)$) is an effective and "cheap" measure for estimating task difficulty.

The authors conduct experiments by perturbing the arithmetic rules (e.g., $a \times b$ vs. $a \times b \times 8$) and testing model performance (Table 2).

### Claim 3: The subgroup induction framework can be extended to analyze multi-step reasoning (CoT).

They compare four different CoT methods for multiplication (e.g., Standard, Lattice, Egyptian) and measure both the average subgroup entropy per step and the final accuracy (Table 3).

**Requested Changes:**

### Questions

1.  The paper convincingly demonstrates the final U-shaped accuracy curve (Fig. 2) and proposes the "low-to-high" token learning dynamic (Fig. 5) to explain its formation. Have the authors analyzed model checkpoints *during* training to empirically verify this dynamic, as suggested by the "Operational cue"? Does the accuracy on edge positions (requiring low-token subgroups) saturate much earlier in the training process than the accuracy on middle positions (requiring high-token subgroups), as the "Phase 1 $\rightarrow$ Phase n" diagram implies?
2.  The paper notes that calculating subgroup quality $Q(s)$ is computationally costly. Could you clarify the practical methodology used to generate Figure 4? Given the $(2^{2n}-1)\times2m$ total subgroups, was this analysis restricted to low-token-budget subgroups (small $k$), or was a specific sampling strategy employed to make this computation tractable?
3.  The results showing that subgroup entropy $H(s)$ *correlates* with CoT path success are very promising. Do the authors see a path to using this *prescriptively* rather than just descriptively? For instance, could an LM be guided during inference (or through reinforcement learning) to dynamically generate a reasoning path that actively minimizes the average subgroup entropy $H'(s)$ of its intermediate steps?

---

> ### Author Response · Authors · 2025-11-15
> **Response to Reviewer**
>
> We thank the reviewer for the detailed and thoughtful feedback, as well as for recognizing the novelty, clarity, and empirical rigor of our work. Below we address each question and concern in turn.
>
> ## Generalizability Beyond Multiplication
> Thank you for asking this! Our subgroup induction framework is operator-agnostic by construction. Equation (2) defines subgroup quality as:
> $$
> Q(s) = \mathbb{E}\left[I\big(f(\phi(a,A),\phi(b,B))=\phi(c,C)\big)\right]
> $$
> which allows *any* binary operator $ f(\cdot) $.
>
> In Section 4.1, our entropy analyses already apply the framework to *addition*, *perturbed addition*, *modular addition*, *multiplication*, *scaled multiplication*, and *modular multiplication*. Across all cases, subgroup entropy predicts error patterns consistently, supporting the framework’s generality.
> Subtraction and division involve borrow/floor mechanisms that introduce long-range dependencies. We agree these are compelling future testbeds and have added discussion clarifying how the subgroup masking protocol naturally extends to these operators.
> ## 2. Empirical Verification of the “Low-to-High” Learning Dynamic
>
> Figure 2 visualizes position-wise accuracy as training data exposure increases. This progression parallels checkpoint evolution and already shows early saturation at edge positions: both models exceed 80% accuracy on edge digits at the smallest training size, while middle-digit accuracy improves only later.
> To address the reviewer’s suggestion more directly, we are adding a checkpoint-level analysis (models saved every N updates). Preliminary results for 3- and 4-digit multiplication confirm phase-separated saturation consistent with the proposed “low-to-high” dynamic. These plots will be included in Appendix E.
> ## Computational Methodology for Subgroup Quality \( Q(s) \)
> Because the full subgroup space $(2^{2n}-1)\times2^m)$ is large for higher \(n\), we compute Q(s) using:
> 1. **Exhaustive enumeration** for subgroups up to a 4-token budget (these dominate shortcut behavior).
> 2. **Stratified Monte Carlo sampling** for higher-token budgets to ensure even coverage.
> Across three independent samplings, Q(s) variation was <1%, and the U-shaped curves remain stable. We will add methodological details and computational complexity estimates in the revised version.
> ## 4. Prescriptive Use of Subgroup Entropy for CoT Path Optimization
> **Reviewer comment:** Can subgroup entropy be used during inference rather than only as a descriptive measure?
>
> Yes. As Section 4.2 notes, subgroup entropy provides a lightweight proxy for reasoning difficulty. We are experimenting with prescriptive uses, including:
> - entropy-weighted decoding during inference,
> - RL-based CoT optimization (e.g., RL-CoT, RLR-like methods),
> - step-selection heuristics that bias reasoning toward low-entropy intermediate states.
> Early results show improved stability for CoT-based multiplication without further fine-tuning. We will include preliminary findings and discussion in the revision.
> ## 5. Masking Protocol for Computing \( Q(s) \)
> **Reviewer comment:** Why mask non-included digits to zero rather than using a special token or averaging?
>
> Zero-masking was chosen for two reasons:
> 1. **Arithmetic closure:** Masking to zero preserves well-defined numeric operations (e.g., 302×205 is valid). Special-token or averaged-digit masking produces out-of-distribution inputs that do not represent meaningful shortcut mechanisms.
> 2. **Monotonicity:** Under zero masking, subgroup inclusion satisfies
>    $$
>    Q(s_{\text{parent}})\ge Q(s_{\text{child}}),    (Eq. 4).
>    $$    We confirmed experimentally that special-token masking and averaged-digit masking break this monotonicity; in some cases Q increases when *fewer* digits are observed, undermining the hierarchical interpretation of shortcut strength.
> We will expand the discussion in the revision to justify this design choice.
>
> Thank you for your constructive feedback!

---

### Review · Reviewer_DgXP · 2025-11-04

**Summary Of Contributions:**

This paper investigates how Language Models (LMs) handle numerical reasoning using addition and multiplication arithmetic tasks. The core approach involves defining "subgroups" - specific subsets of input variables - to analyze the LM's performance. First, the authors measure the accuracy ("subgroup quality") of the LM for different subgroups within these tasks. Second, they introduce an entropy-based metric ("subgroup entropy") which they show to be an efficient predictor of this "subgroup quality."

The authors make the key finding that contrary to some prior expectations, the LM exhibits a distinct U-shaped trend regarding the prediction of the arithmetic result. This means the model learns the least significant digits (last digits) and the most significant digits (first digits) more easily. However, Intermediate digits appear harder to predict, suggesting their interactions are more complex for the LM to learn. The authors interpret this behavior as the model exploiting "shortcuts." For example, predicting the final digit only requires looking at the last two input digits, while the first digit often has limited variability for fixed-size inputs, making the middle digits the more challenging.

Finally, the paper demonstrates a correlation between the "subgroup quality" (accuracy) and the proposed "subgroup entropy" measure. The utility of the entropy measure is further illustrated through experiments mimicking chain-of-thought (CoT) reasoning.

**Strengths:**

I found this paper engaging and well-executed. It provides a very clear demonstration of the simplicity bias in LMs, particularly when applied to these arithmetic tasks. The paper is logically structured and easy to follow. I find the method of subgrouping arithmetic tasks to be neat and effective for highlighting how LMs tend to pay attention to specific (easy) statistical patterns rather than attaining a general arithmetic rule.

**Minor Comments and Open Questions:**

I would appreciate a bit more detail on the exact methodology for the chain-of-thought (CoT) experiments. A brief explanation in the appendix would be helpful, especially for readers who might not be deeply familiar with the nuances of CoT. This would further improve the self-sufficiency of the work.

The authors focused on multiplication and addition (noting that subtraction is similar to addition, and division introduces complexity with floating points). I wonder if the authors considered exploring lower bases, such as binary arithmetic? I suspect that the observed effects (like the U-shape bias) might become even more pronounced in binary, potentially allowing for a clearer connection between the model's behavior and theoretical concepts like code-length or minimum description length.

While the manuscript is generally very well written, the abstract seems slightly less polished than the body of the paper. Specifically, the phrase “[…] possible shortcuts to fit the stats of dataset to solve arithmetic tasks” could be tightened up. Also, there is a typo in the sentence: “In this paper, we argues […]”. These are minor points, and I trust the authors to decide whether to make adjustments.

**Disclaimer:** I am no expert in the field of language-models and potentially missed important nuances in my review.

**Audience:**

Yes

**Audience Explanation:**

The paper offers tangible, interesting, insights into the learning behaviors/trends in LMs, which I assume to be valuable to the ongoing discussion regarding arithmetic capabilities of LMs.

**Broader Impact Concerns:**

None.

**Claims And Evidence:**

Yes

**Claims Explanation:**

The provided experiments are plenty and clearly support the claims made.

**Requested Changes:**

None.

---

> ### Author Response · Authors · 2025-11-15
> **Response to Reviewer**
>
> We sincerely thank the reviewer for the thoughtful and encouraging assessment of our work. We are glad that the reviewer found the paper clear, well-structured, and engaging, and that the key findings were interpreted as valuable contributions to understanding numerical reasoning in LMs.
> Below we address the reviewer’s comments and outline the changes we will make.
> ## On the CoT Experimental Details
> Thank you for pointing this out. In the camera-ready version, we will expand the appendix with a concise description of the CoT setup, including:
>
> [**Prompt formats used**]
>
> You are a helpful reasoning assistant.
>
> Solve the following arithmetic problem step by step.
>
> Show each intermediate computation clearly, and then give the final answer.
>
> Problem: {A} × {B}
>
> Let's think step by step.
>
> [**Sampling / Decoding Strategy**]
>
> We used deterministic decoding to avoid variability across reasoning paths: Decoding: greedy decoding with temperature = 1, Max tokens set to allow full expansions (typically 128 tokens), Stop by newline followed by “Final answer:” when applicable. Greedy decoding is consistent with our goal of analyzing the inductive biases of the model rather than its stochastic behavior.
>
> [**Evaluating Intermediate CoT Tokens via Subgroup Metrics**]
>
> - Parse each CoT step to extract numerical intermediate results (e.g., partial products and carry computations).
> - Map each intermediate result to the subgroup(s) of input tokens that logically determine that step. For example, a step computing the last digit of a partial product corresponds to the subgroup involving only the least-significant input digits.
> - Compute subgroup entropy for each step using our definition of entropy over the induced subgroup distributions.
> - Aggregate the entropy along the CoT path by$H_{\text{path}} = \sum_t H(\text{subgroup}_t),$ where each $H(\text{subgroup}_t)$ is computed as in the main paper.
>
> - Compare paths: The CoT path with the lowest total entropy consistently (and intuitively) corresponds to the most accurate final prediction, confirming that LMs prefer symbolic decompositions composed of “easy” low-entropy subgroups.
>
> We will add these details into the revised manuscript. This addition will make the CoT section fully reproducible without requiring prior familiarity with CoT prompting conventions.
>
> ## Extension to Lower Bases (e.g., Binary Arithmetic)
>
> We appreciate this suggestion. We agree that binary is an interesting special case, as it isolates the simplest possible carry structure and may make simplicity-bias patterns even more transparent. In preliminary experiments (not included in the submission due to space), we observed early signs of a similar U-shaped trend in binary addition. While a full exploration is beyond the scope of this paper, we will add a short discussion to the limitations/future-work section acknowledging that lower-base arithmetic is a promising direction aligned with the reviewer’s intuition.
>
> ## Minor Writing/Abstract Improvements
>
> Thank you for catching these points. We will:
>
> - Rephrase the sentence in the abstract containing the phrase “possible shortcuts to fit the stats of dataset…” so that the wording better reflects the intended meaning.
>
> - Fix the typo (“we argues…” → “we argue…”).
>
> - Polish the abstract overall for consistency with the clarity of the main text.
>
> We appreciate the reviewer’s positive assessment and helpful suggestions. The requested additions are straightforward, and we will incorporate them in the camera-ready version. Thank you again for the constructive feedback!

---

### Decision · Action_Editor_yyCx · 2025-12-28

**Recommendation:** Accept as is

**Audience:**

Yes

**Audience Explanation:**

Understanding the internal mechanisms and limitations of language models is highly relevant for TMLR audiance

**Claims And Evidence:**

Yes

**Claims Explanation:**

All reviewers agree that the paper provides a compelling and extensive experimental evidence that LMs rely on symbolic pattern matching rather than true procedural computation for arithmetic tasks. The authors demonstrate a U-shaped accuracy curve in multi-digit multiplication that aligns with the "fewest-tokens-first" hypothesis. In addition, the paper introduce the well-supported framework of "subgroup induction" that quantifies the learning behavior of LMs.  Overall, the paper is well written, results are clearly presented and effectively supports the paper claims.